# Hierarchical Synthetic Tabular Data Generation: A Hybrid Top-Down and Bottom-Up Framework

**Junfeng Nie** [1 2]   **Alvin Jin** [1 2]   **Xiaohui Chen** [1 2]

## Abstract

Existing approaches for synthetic tabular data generation are based on either purely generative models or LLMs, both of which struggle with data heterogeneity, logical consistency, rare-event coverage, and robustness in low-data regimes. In this paper, we propose a hierarchical hybrid top-down and bottom-up (H-TDBU) framework that decouples semantic structures from stochastic texture. In the top-down path, structure-driven logical constraints and cross-modal alignment rules are constructed, while in the bottom-up path, lightweight tabular generators are used to learn local statistical patterns from real data. The two paths are consolidated in a unified synthesis engine with an iterative feedback loop. We evaluate the framework on weak multimodal financial benchmarks combining tabular and sentiment-text data. Experimental results show that our H-TDBU approach improves train-synthetic-test-real performance over neural baseline methods while preserving semantic consistency. Our results suggest that hierarchical rule-guided synthesis provides an effective mechanism for combining controllability, semantic coherence, and statistical fidelity in synthetic data generation.

## 1. Introduction

Tabular data remains a dominant information structure for decision-making systems in domains such as finance, healthcare, and policy. However, its effective use in modern machine learning pipelines is bottlenecked by a combination of scattered data sources, structural heterogeneity, and privacy constraints (Shi et al., 2025; Manousakas & Aydöre, 2023). Synthetic data generation has emerged as a promising mechanism to mitigate these limitations by producing artificial datasets with new information and structural compliance (Byun et al., 2025; Wang et al., 2024).

A large body of existing work has been focused on learning the distribution of real-world data, including CTGAN and TVAE (Xu et al., 2019; Zhao et al., 2023), as well as more recent tabular diffusion models such as STaSy (Kim et al., 2023), SOS (Kim et al., 2022) and TabDDPM (Kotelnikov et al., 2023). These methods approximate the distribution of tabular data and sample new data points from the learned distribution. While effective in certain benchmarks, they require substantial amounts of training data and struggle with heterogeneous feature types, leading to potential *mode collapse* (Shumailov et al., 2024). This means that unconditional generative models tend to miss important rare and tail events that are crucial in financial applications such as fraud detection. In low-data regimes, which are a key motivating scenario for using synthetic data, these methods tend to overfit and produce lower diversity samples.

There is another line of recent work exploring the in-context learning capability of large language models (LLMs). Methods such as GReaT (Borisov et al., 2023), REaLTab-Former (Solatorio & Dupriez, 2023), and TabuLa (Zhao et al., 2025) re-structure table rows as strings and use text foundation models to generate synthetic data points. While such methods benefit from zero-shot synthesis and semantic priors embedded in foundation models, they face several limitations. Notably, autoregressive token generation does not enforce structural consistency, logical constraints, or functional dependencies among attributes, often leading to invalid or incoherent samples. Moreover, sequential row tokenization weakens the modeling of heterogeneous numerical-categorical interactions and long-range feature dependencies. Such methods also struggle to adequately capture tail behaviors and combinatorial edge cases, resulting in limited semantic coverage and controllability.

A deeper challenge, largely overlooked in many prior works, is that synthetic data generation is often treated as a purely generative problem. In reality, enterprise-scale data are highly structured, heterogeneous, and distributed across multiple sources and modalities, creating substantial integration and consistency challenges (Putrama & Martinek,

---

[1]AnyFluxion [2]University of Southern California. Correspondence to: Nie Junfeng <junfeng.nie@anyfluxion.com>, Alvin Jin <alvin.jin@anyfluxion.com>, Xiaohui Chen <xiaohui.chen@anyfluxion.com>.

*Proceedings of the 43rd International Conference on Machine Learning*, Seoul, South Korea. PMLR 306, 2026. Copyright 2026 by the author(s).

2024). Existing methods either ignore this multi-modality or attempt to unify it within a single umbrella model. One key issue of those approaches is that the sole reliance on model-generated data introduces the risk of mode/distribution collapse, where recursively training on synthetic outputs progressively wipes out the underlying data distribution, particularly in the tails (Shi et al., 2025). To mitigate this issue, a recent line of work proposed a multi-stage controllable synthetic data generation approach based on the reasoning capability of LLMs (Davidson et al., 2026; Umesh et al., 2026). The central idea is a hierarchical semantic decomposition into a general structure that explicitly defines the "coverage space" of the data set. However, their approach heavily relies on LLM reasoning quality and is operationally expensive in terms of many LLM calls.

In this work, we organically integrate the generative and hierarchical perspectives. In the top-down path, we define structure rules via either manual queries or LLM prompts. To complement the structure generation, we couple a lightweight (i.e., low-compute) tabular data generative model (e.g., ensemble methods) in a way that the latent space representation aligns with the generated structure rules in the top-down stage. The top-down and bottom-up paths are reconciled in a unified synthesis engine. Through a feedback loop validating cross-modal metrics, we can maintain logical consistency and high realism. Figure 1 shows a diagram that illustrates our proposed framework. We emphasize that LLMs are only used in the top-down path to generate the scheme representing the structure rules. The actual tabular synthetic data are generated in the bottom-up path by training much cheaper agent models, such as random forests (Breiman, 2001) or XGBoost (Chen & Guestrin, 2016). Code, configurations, and reproducibility scripts are available in our public GitHub repository.

## 2. Our Method

We introduce a robust framework that combines hybrid top-down and bottom-up synthesis with a hierarchical top-down rule construction process.

**Top-Down: Structure.** Based on human instructions or LLM prompts, we define a structural template $\mathcal{S}$ that acts as the backbone of the synthetic data via learning factors $f_i$ that define the structure of the data. This top-down approach yields domain expertise. Here, the rules of the dataset are established, but the dataset lacks realism. The key output of this step is the logical framework $\mathcal{S}$, which serves as a structured map of a concept space, breaking down abstract factors into concrete, sampleable attributes.

**Bottom-Up: Texture.** To complement the top-down approach, we start with a sample of the real data $\mathcal{D}_{\text{real}}$. These data serve as the template from which ensemble methods

and generative methods (e.g., CTGAN) learn. We denote these base generative methods as $G$. The output of this process yields a latent space $\mathcal{Z}$ that captures detailed complex patterns that look like real data, but may not follow the logical rules.

**Synthesis and Reconciliation.** Both the logic skeleton $\mathcal{S}$ and the latent texture $z$ are fed into the hybrid top-down/bottom-up generator $X_{\text{Syn}} := G(z \in \mathcal{Z}|\mathcal{S})$ to generate data from the latent noise consistent with the logical constraints of $\mathcal{S}$. The conditional generator here reconciles the logic skeleton $\mathcal{S}$ with the latent texture $z$ derived from the base model. The output is $\mathcal{D}_{\text{syn}}$, which combines high realism with compliance by construction. We evaluate via training on synthetic data and testing on real data. If the synthetic data $\mathcal{D}_{syn}$ falls below a threshold for XModal (cross-modal fidelity), the system triggers a Retrain Model signal; if it fails Constraint Validation, it triggers an Adjust Constraints signal to the Top-Down provider.

## 3. Empirical Results

**Experimental Setting.** We evaluate the workflow on four benchmark datasets: the manual weak multimodal benchmark, the Gemini-generated weak multimodal benchmark, Adult Income (Turanyksel, 2021), and German Credit (Ahmedtronic, 2021). Both weak multimodal benchmarks are constructed from Bank Marketing (Moro et al., 2014) as the tabular source and FinancialPhrase-Bank financial-news sentiment data (Zing, 2020) as the text source. The manual benchmark uses a hand-written JSON rule provider that maps target 1 to positive or neutral financial text and target 0 to neutral or negative financial text. The Gemini benchmark uses *Gemini 3.1 Pro* to generate the rule-provider JSON from a compact dataset summary, including the Bank Marketing target distribution and the FinancialPhraseBank sentiment-label distribution. In the Gemini-generated rule, target 1 is aligned only with positive text, while target 0 is aligned with neutral or negative text. This design isolates the role of the LLM as a top-down rule provider: the LLM changes the alignment rules, while the bottom-up synthesis methods and evaluation pipeline remain fixed.

For each dataset and synthesis method, we generate 12,000 synthetic rows. For the low-compute synthesis methods, we repeat experiments over three random seeds, 42, 123, and 2024, covering independent column sampling, Gaussian copula, RandomForest conditional synthesis, and XGBoost conditional synthesis. For the RandomForest generator, we use at most 5,000 training rows, 30 trees, and at most 12 conditioning columns. For the XGBoost generator, we use at most 8,000 training rows, at most 12 conditioning columns, 80 estimators, and maximum tree depth 6. The fixed synthetic sample size and repeated seeds make the comparison

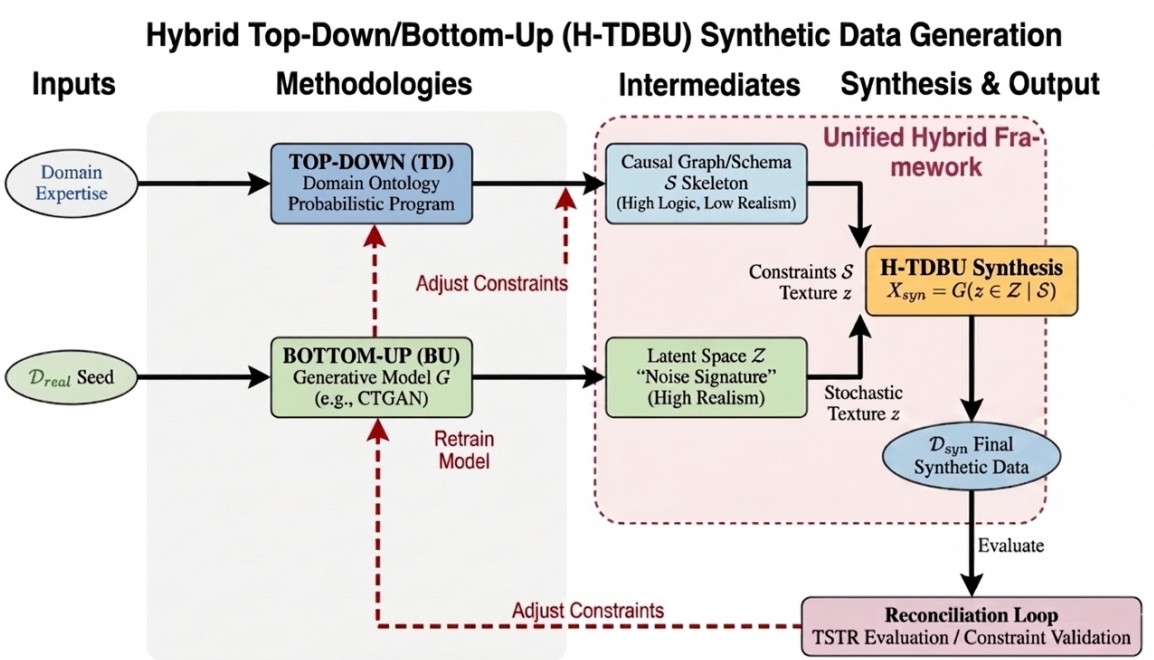

*Figure 1.* The unified Hybrid Top-Down/Bottom-Up (H-TDBU) framework. The Top-Down stream establishes logical constraints $\mathcal{S}$, while the Bottom-Up stream learns latent stochastic texture $z$. The two paths are reconciled in a unified synthesis engine, $G(z \in \mathcal{Z} \mid \mathcal{S})$. The reconciliation loop validates the output against TSTR and cross-modal metrics (e.g., XModal) to provide feedback for refining either the Top-Down constraints or the Bottom-Up model, preventing model drift and ensuring logical integrity.

reproducible and reduce sensitivity to sampling randomness across methods. We also run SDV neural baselines, CTGAN and TVAE, with 50 training epochs and seed 42. These SDV results provide neural baseline comparisons, but are treated as preliminary comparison points rather than repeated-seed estimates because of their higher computational cost.

For the XGBoost ablation, we vary the amount of training data and the number of conditioning columns while keeping the same evaluation protocol. The ablation settings are: 1,000 training rows, 3,000 training rows, 4 conditioning columns, 8 conditioning columns, and 12 conditioning columns. Each ablation again generates 12,000 synthetic rows and is repeated over seeds 42, 123, and 2024. The default XGBoost setting corresponds to 8,000 training rows, 12 conditioning columns, 80 estimators, and maximum depth 6. This ablation tests whether XGBoost performance is driven mainly by more training rows or by the number of available conditioning columns.

**Evaluation Metrics.** We evaluate synthetic data using downstream utility and fidelity. For downstream utility, we use train-synthetic-test-real (TSTR), where a logistic regression classifier is trained on synthetic data and evaluated on held-out real data; the Acc., F1, and AUROC columns in Tables 1 and 2 report these TSTR metrics. We also compute the gap between TSTR and train-real-test-real (TRTR), where TRTR is the same classifier trained and evaluated

using real data; this gap reported in Appendix A is used as a diagnostic metric. Smaller absolute gaps indicate that synthetic data better preserves downstream predictive structure.

For fidelity, we compare real and synthetic marginal statistics. Numeric fidelity is measured by the average absolute difference in column means and standard deviations. Categorical fidelity is measured by average total variation distance across categorical columns. For weak multimodal benchmarks, we also report a cross-modal difference metric, denoted as XModal in Tables 1 and 2. This metric measures the discrepancy between the real and synthetic joint distributions of the tabular target and attached text sentiment. Lower values indicate better preservation of the weak cross-modal alignment.

**Results Analysis.** Tables 1 and 2 summarize the weak multimodal results. Figure 2 visualizes three complementary quantities: TSTR AUROC, TSTR F1, and XModal. Accuracy is reported in the tables, while the figure emphasizes AUROC and F1 because the benchmarks are class-imbalanced; a method can obtain high accuracy by predicting the majority class while still failing to learn the minority positive class.

On the manually specified benchmark, independent column sampling performs poorly in downstream utility, achieving 0.4905 AUROC and 0.0000 F1. This confirms that inde-

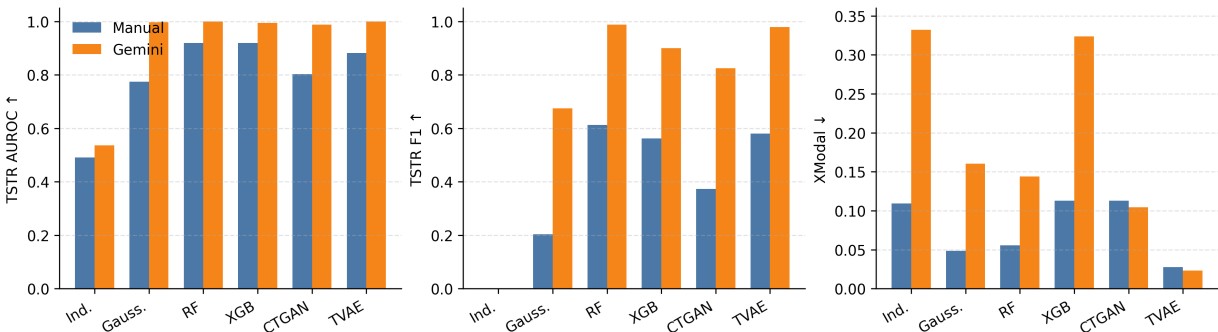

*Figure 2.* Weak multimodal comparison across synthesis methods. Panels show TSTR AUROC, TSTR F1, and XModal; higher AUROC/F1 and lower XModal are better.

pendently sampling columns does not preserve the predictive dependencies needed for downstream learning. Conditional generators perform substantially better. RandomForest reaches 0.9281 accuracy, 0.6122 F1, and 0.9188 AUROC, while XGBoost reaches 0.9139 accuracy, 0.5621 F1, and 0.9190 AUROC. Among neural baselines, TVAE achieves 0.8827 AUROC and 0.5581 F1, while CTGAN reaches 0.8358 AUROC and 0.2694 F1. Thus, in this setting, the lower-compute tree-based conditional methods are competitive with or stronger than the neural baselines.

The LLM-generated rule provider produces a stricter top-down alignment because target 1 is aligned only with positive financial text. Under this setting, Figure 2 shows that most conditional and neural methods improve over independent sampling in both AUROC and F1. RandomForest obtains the strongest downstream utility, with 0.9971 accuracy, 0.9878 F1, and 0.9998 AUROC. XGBoost also performs strongly, reaching 0.9746 accuracy, 0.9003 F1, and 0.9948 AUROC. CTGAN and TVAE achieve 0.9903 and 0.9885 AUROC, respectively, showing that neural baselines can also exploit this more separable alignment structure.

Cross-modal fidelity shows a different trade-off. On the manual benchmark, Gaussian copula has the lowest XModal value, 0.0485, followed by TVAE at 0.0533 and RandomForest at 0.0555. On the Gemini-generated benchmark, TVAE has the lowest XModal value, 0.0193, while RandomForest has stronger downstream utility but a larger XModal value of 0.1437. This suggests that high downstream utility and

*Table 1.* Manual weak multimodal benchmark results. Method results are averaged over seeds 42, 123, and 2024. CTGAN and TVAE use seed 42. Arrows indicate metric direction; bold indicates the best value in the table.

| Method | Acc. ↑ | F1 ↑ | AUROC ↑ | XModal ↓ |
|---|---|---|---|---|
| Independent | 0.8830 | 0.0000 | 0.4905 | 0.1094 |
| Gaussian | 0.8949 | 0.2034 | 0.7738 | **0.0485** |
| RandomForest | **0.9281** | **0.6122** | 0.9188 | 0.0555 |
| XGBoost | 0.9139 | 0.5621 | **0.9190** | 0.1127 |
| CTGAN | 0.8992 | 0.2694 | 0.8358 | 0.0646 |
| TVAE | 0.8622 | 0.5581 | 0.8827 | 0.0533 |

*Table 2.* Gemini-generated weak multimodal benchmark results.

| Method | Acc. ↑ | F1 ↑ | AUROC ↑ | XModal ↓ |
|---|---|---|---|---|
| Independent | 0.8830 | 0.0000 | 0.5359 | 0.3320 |
| Gaussian | 0.9423 | 0.6749 | 0.9968 | 0.1605 |
| RandomForest | **0.9971** | **0.9878** | **0.9998** | 0.1437 |
| XGBoost | 0.9746 | 0.9003 | 0.9948 | 0.3234 |
| CTGAN | 0.9574 | 0.7863 | 0.9903 | 0.1225 |
| TVAE | 0.9881 | 0.9476 | 0.9885 | **0.0193** |

low cross-modal discrepancy are related but not identical objectives.

The tabular-only benchmarks provide an additional check that method performance depends on dataset structure. On Adult Income, RandomForest obtains the best AUROC among the fast methods, reaching 0.8770. On German Credit, Gaussian copula performs best with 0.7750 AUROC. This variation suggests that the workflow should be interpreted as a benchmark framework rather than a claim that one generator dominates across all datasets. Additional TRTR/TSTR gap, tabular-only, fidelity, and ablation results are reported in Appendix A.

Table 3 summarizes the XGBoost ablation. On the manual benchmark, the best AUROC is obtained with 12 conditioning columns, while on the Gemini-generated benchmark the best AUROC is obtained with 4 conditioning columns. This is consistent with the structure of the two rule providers: the manual rule is more permissive because neutral text can appear with both target values, while the Gemini-generated rule creates a stricter alignment where target 1 is paired only with positive text. As a result, the Gemini benchmark has fewer conditioning columns, whereas the manual benchmark benefits from richer conditioning context.

Overall, the results support the hybrid top-down / bottom-up framing: top-down rules define the weak multimodal

*Table 3.* XGBoost ablation summary. The best ablation is selected by AUROC.

| Benchmark | Best setting | Acc. ↑ | F1 ↑ | AUROC ↑ |
|---|---|---|---|---|
| Manual | 12 cols | 0.9139 | 0.5621 | 0.9190 |
| Gemini | 4 cols | **0.9925** | **0.9690** | **0.9999** |

structure, while bottom-up generators determine how well that structure is preserved in synthetic data. In particular, the Gemini-generated rule provider shows that an LLM can be used at the rule-construction stage without directly generating synthetic rows.

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

## A. Additional Empirical Results

This appendix provides additional utility, fidelity, and ablation results supporting the empirical analysis in the main text.

### A.1. TRTR, TSTR, and Gap Diagnostics

The main paper reports TSTR utility metrics in Tables 1 and 2. For completeness, Appendix Table 4 reports the corresponding TRTR reference, TSTR value, and gap for accuracy, F1, and AUROC. Each gap is computed as TSTR minus TRTR for the same metric, so values closer to zero indicate that training on synthetic data approaches the performance of training on real data. The empirical gaps are reported in Figure 3.

*Table 4.* Weak multimodal utility diagnostics. Gap is computed as TSTR minus TRTR.

| Benchmark | Method | TRTR Acc. ↑ | TSTR Acc. ↑ | Gap Acc. ↑ | TRTR F1 ↑ | TSTR F1 ↑ | Gap F1 ↑ | TRTR AUROC ↑ | TSTR AUROC ↑ | Gap AUROC ↑ |
|---|---|---|---|---|---|---|---|---|---|---|
| Manual | Independent | 0.9308 | 0.8830 | -0.0478 | 0.6437 | 0.0000 | -0.6437 | **0.9460** | 0.4905 | -0.4555 |
| Manual | Gaussian | 0.9308 | 0.8949 | -0.0359 | 0.6437 | 0.2034 | -0.4403 | **0.9460** | 0.7738 | -0.1722 |
| Manual | RandomForest | 0.9308 | **0.9281** | **-0.0028** | 0.6437 | **0.6122** | **-0.0315** | **0.9460** | 0.9188 | -0.0272 |
| Manual | XGBoost | 0.9308 | 0.9139 | -0.0169 | 0.6437 | 0.5621 | -0.0816 | **0.9460** | 0.9190 | **-0.0270** |
| Manual | CTGAN | **0.9313** | 0.8971 | -0.0342 | **0.6450** | 0.3737 | -0.2713 | 0.9449 | 0.8034 | -0.1415 |
| Manual | TVAE | **0.9313** | 0.8932 | -0.0381 | **0.6450** | 0.5802 | -0.0648 | 0.9449 | 0.8815 | -0.0634 |
| Gemini | Independent | **1.0000** | 0.8830 | -0.1170 | **1.0000** | 0.0000 | -1.0000 | **1.0000** | 0.5359 | -0.4641 |
| Gemini | Gaussian | **1.0000** | 0.9423 | -0.0577 | **1.0000** | 0.6749 | -0.3251 | **1.0000** | 0.9968 | -0.0032 |
| Gemini | RandomForest | **1.0000** | **0.9971** | **-0.0029** | **1.0000** | 0.9878 | -0.0122 | **1.0000** | **0.9998** | **-0.0002** |
| Gemini | XGBoost | **1.0000** | 0.9745 | -0.0255 | **1.0000** | 0.9003 | -0.0997 | **1.0000** | 0.9948 | -0.0052 |
| Gemini | CTGAN | **1.0000** | 0.9620 | -0.0380 | **1.0000** | 0.8248 | -0.1752 | **1.0000** | 0.9882 | -0.0118 |
| Gemini | TVAE | **1.0000** | 0.9948 | -0.0052 | **1.0000** | 0.9782 | -0.0218 | **1.0000** | 0.9995 | -0.0005 |

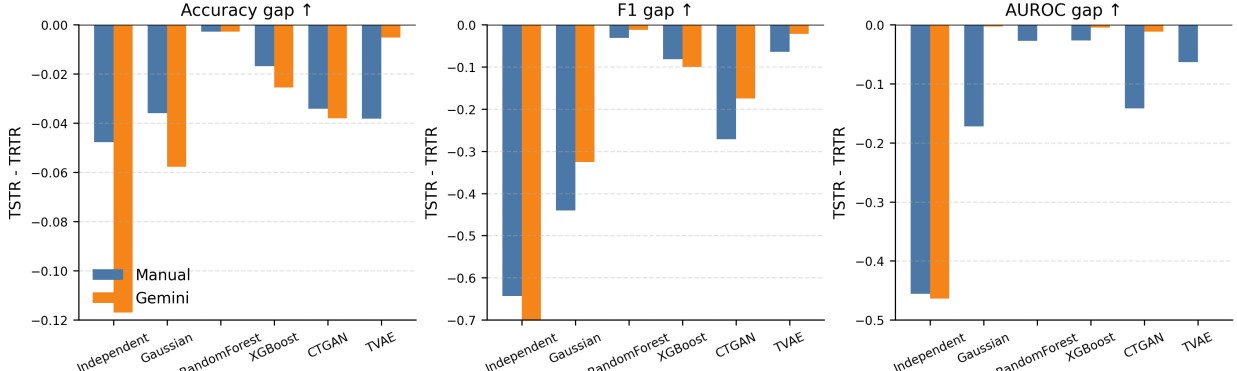

*Figure 3.* Weak multimodal utility gaps. Bars closer to zero indicate smaller degradation relative to the real-data reference.

### A.2. Tabular-Only Benchmark Results

Table 5 reports additional results for the tabular-only Adult Income and German Credit benchmarks. These results are used as a robustness check because they remove the weak text-tabular alignment layer and evaluate whether the same synthesis pipeline behaves reasonably on standard tabular datasets.

*Table 5.* Additional tabular-only benchmark results for Adult Income and German Credit.

| Dataset | Method | TRTR AUROC ↑ | TSTR AUROC ↑ | Gap AUROC ↑ | TSTR Acc. ↑ | TSTR F1 ↑ |
|---|---|---|---|---|---|---|
| Adult | Independent | **0.9063** | 0.5667 | -0.3396 | 0.7600 | 0.0020 |
| Adult | Gaussian | **0.9063** | 0.8023 | -0.1040 | 0.8057 | 0.4309 |
| Adult | RandomForest | **0.9063** | **0.8770** | **-0.0293** | 0.8098 | 0.3914 |
| Adult | XGBoost | **0.9063** | 0.8400 | -0.0663 | 0.7961 | 0.2690 |
| Adult | CTGAN | 0.9061 | 0.8699 | -0.0362 | **0.8235** | 0.5939 |
| Adult | TVAE | 0.9061 | 0.8729 | -0.0331 | 0.8036 | **0.6548** |
| German | Independent | 0.7833 | 0.5069 | -0.2764 | 0.7000 | 0.8235 |
| German | Gaussian | 0.7833 | **0.7750** | **-0.0083** | **0.7253** | **0.8336** |
| German | RandomForest | 0.7833 | 0.5921 | -0.1912 | 0.7000 | 0.8235 |
| German | XGBoost | 0.7833 | 0.5879 | -0.1953 | 0.6133 | 0.6978 |
| German | CTGAN | **0.7955** | 0.3883 | -0.4072 | 0.7000 | 0.8235 |
| German | TVAE | **0.7955** | 0.5000 | -0.2955 | 0.7000 | 0.8235 |

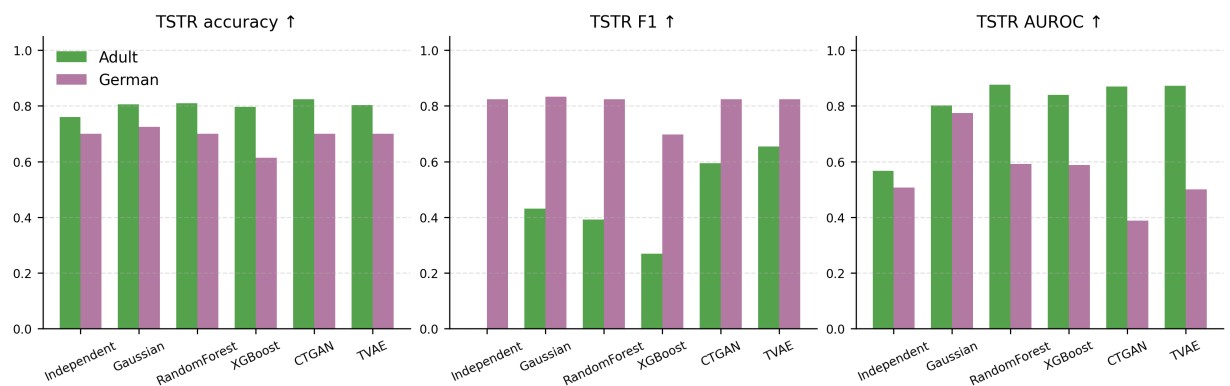

*Figure 4.* Tabular-only TSTR utility comparison on Adult Income and German Credit. Higher accuracy, F1, and AUROC indicate better downstream utility.

## A.3. Fidelity Diagnostics

Table 6 reports additional fidelity metrics for all benchmarks and methods. Figure 5 visualizes the categorical and cross-modal fidelity metrics on the weak multimodal benchmarks.

*Table 6.* Additional fidelity diagnostics. Numeric columns report average absolute differences in means and standard deviations; categorical TVD is the average total variation distance across categorical columns; XModal is reported only for weak multimodal benchmarks.

| Dataset | Method | Num. mean diff ↓ | Num. std diff ↓ | Cat. TVD ↓ | XModal ↓ |
|---|---|---|---|---|---|
| Adult | Independent | 152.3788 | 517.2606 | **0.0066** | – |
| Adult | Gaussian | **57.3640** | **313.2183** | 0.0069 | – |
| Adult | RandomForest | 140.1073 | 587.0696 | 0.0697 | – |
| Adult | XGBoost | 341.9150 | 867.1245 | 0.1286 | – |
| Adult | CTGAN | 1717.9532 | 914.2598 | 0.0806 | – |
| Adult | TVAE | 14040.7644 | 14116.1852 | 0.1012 | – |
| German | Independent | 9.8224 | 9.0639 | 0.0050 | – |
| German | Gaussian | **6.2458** | **7.2130** | **0.0048** | – |
| German | RandomForest | 160.8861 | 12.0561 | 0.1078 | – |
| German | XGBoost | 73.6746 | 171.2152 | 0.1226 | – |
| German | CTGAN | 868.5763 | 708.1753 | 0.0569 | – |
| German | TVAE | 622.2928 | 798.8115 | 0.3882 | – |
| Manual | Independent | **0.6957** | 3.5714 | 0.0044 | 0.1094 |
| Manual | Gaussian | 0.8784 | **1.5979** | **0.0040** | 0.0485 |
| Manual | RandomForest | 1.4936 | 9.7587 | 0.0522 | 0.0555 |
| Manual | XGBoost | 1.4599 | 9.7725 | 0.0531 | 0.1127 |
| Manual | CTGAN | 1.7483 | 5.2539 | 0.0790 | 0.1128 |
| Manual | TVAE | 26.4238 | 68.0696 | 0.0908 | **0.0276** |
| Gemini | Independent | 0.6963 | 3.5686 | 0.0047 | 0.3320 |
| Gemini | Gaussian | **0.6229** | **1.7612** | **0.0035** | 0.1605 |
| Gemini | RandomForest | 1.5259 | 9.7661 | 0.0543 | 0.1437 |
| Gemini | XGBoost | 1.4502 | 9.7713 | 0.0644 | 0.3234 |
| Gemini | CTGAN | 6.1461 | 8.6506 | 0.0735 | 0.1045 |
| Gemini | TVAE | 26.8123 | 69.0225 | 0.1301 | **0.0233** |

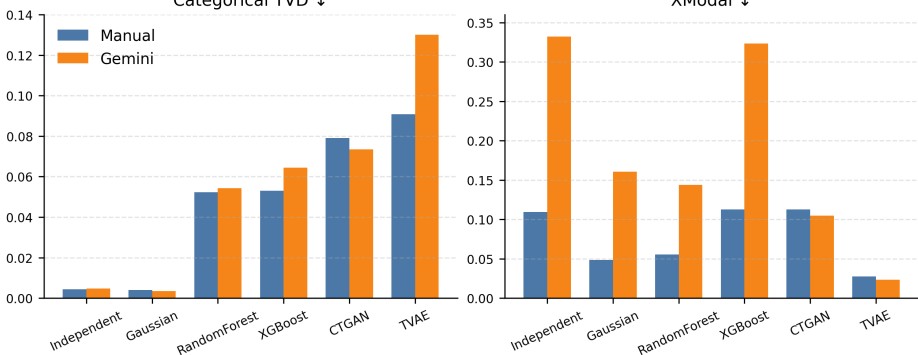

*Figure 5.* Weak multimodal fidelity comparison. Lower categorical TVD and lower XModal indicate closer agreement between real and synthetic data.

### A.4. Full XGBoost Ablation

Table 7 gives the full XGBoost ablation used to produce the summary in Table 3. The ablation varies either the number of training rows or the number of conditioning columns while keeping the remaining evaluation protocol fixed.

*Table 7.* Full XGBoost ablation results. Training-row ablations use the default conditioning setting; conditioning-column ablations use the default training-row setting.

| Benchmark | Setting | TRTR AUROC ↑ | TSTR AUROC ↑ | Gap AUROC ↑ | TSTR Acc. ↑ | TSTR F1 ↑ |
|---|---|---|---|---|---|---|
| Manual | 1k rows | **0.9460** | 0.8985 | -0.0475 | 0.9044 | 0.5404 |
| Manual | 3k rows | **0.9460** | 0.9105 | -0.0355 | 0.9067 | 0.5363 |
| Manual | 4 cols | **0.9460** | 0.7749 | -0.1711 | 0.9009 | 0.3400 |
| Manual | 8 cols | **0.9460** | 0.9162 | -0.0298 | 0.9124 | 0.5389 |
| Manual | 12 cols | **0.9460** | **0.9190** | **-0.0270** | **0.9139** | **0.5621** |
| Gemini | 1k rows | **1.0000** | 0.9866 | -0.0134 | 0.9581 | 0.8376 |
| Gemini | 3k rows | **1.0000** | 0.9922 | -0.0078 | 0.9691 | 0.8797 |
| Gemini | 4 cols | **1.0000** | **0.9999** | **-0.0001** | **0.9925** | **0.9690** |
| Gemini | 8 cols | **1.0000** | 0.9954 | -0.0046 | 0.9766 | 0.9079 |
| Gemini | 12 cols | **1.0000** | 0.9948 | -0.0052 | 0.9745 | 0.9003 |

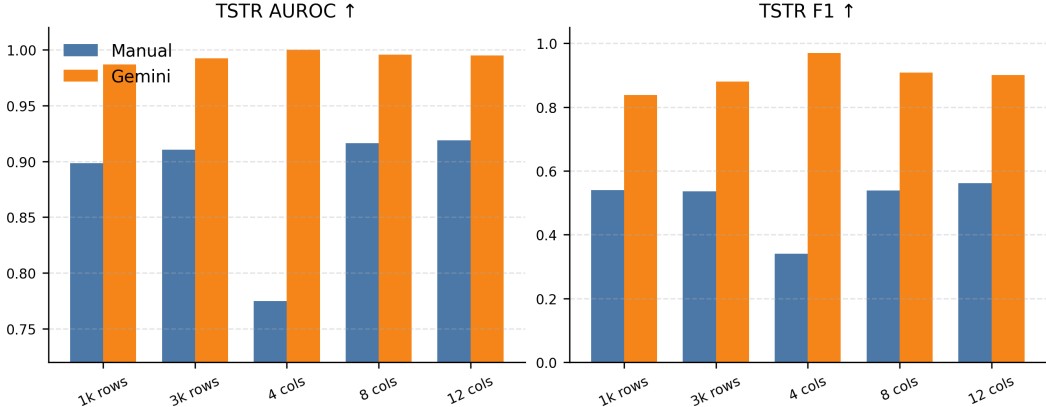

*Figure 6.* XGBoost ablation comparison. Training-row settings vary the amount of real data used to fit the generator, while conditioning-column settings vary the number of previous columns used during sequential conditional synthesis.

