# OpenReview forum: "Hierarchical Synthetic Tabular Data Generation: A Hybrid Top-Down and Bottom-Up Framework"
_ICML.cc/2026/Workshop/FMSD — FMSD @ ICML 2026 Poster_

### Official Review · Reviewer_Wefo · 2026-05-20
**Interesting approach for synthetic tabular data generation that combines two methods**

**Rating:** 9
**Confidence:** 4

**Review:**

Summary of Contributions
H-TDBU decouples synthetic tabular data generation into a top-down path (logical constraints via manual rules or LLM-generated schemas) and a bottom-up path (lightweight generators learning statistical patterns), reconciled via an iterative feedback loop. Evaluated on weak-multimodal financial benchmarks and standard tabular datasets, it is able to preserve model performance and the dataset utility well.

Strengths
- Clear, intuitive method explanation.
- Strong ablations isolating key design choices (training rows, conditioning columns).
- Effective visualizations comparing methods across utility and fidelity.
- Addresses a practical gap

Weaknesses
- Could use more multi-modal benchmarks (both multimodal benchmarks derive from Bank Marketing).

Suggestions
- Broaden evaluation to additional domains.

---

### Official Review · Reviewer_mqxi · 2026-05-21

**Rating:** 4
**Confidence:** 3

**Review:**

## Summary

This paper proposes a method for multimodal tabular generation combining top-down rules-based logic with bottom-up distribution learning. The two distinct paths are synthesized before generation is performed.

## Strengths

- Good motivation: the low data regime is where generation is most needed, but is typically the most underserved regime by generators
- Novel idea to combine a top-down and bottom-up generator

## Areas of Improvement

- Paper is fundamentally not very well-written and is pretty difficult to understand. Crucial details are missing in the main text and impact the flow / readability of the work. For example:
  - What does $\mathcal S$ look like practically?
  - What is the structure of $G$?
  - How is the training done in detail? Loss functions, etc?
- It's not clear to me that the training data is not just mostly memorized in this model. How does this method compare to a simple generator like SMOTE?
- It's not clear to me that the textual information is useful.

## Detailed Comments

- Various typos are in the work
- What is SDV?
- Feels weird to use TVAE or CTGAN when there are way better tabular generators available.

## Justification of Score

While this paper is in a topic that's interesting to the workshop, I believe it significantly misses the mark in writing and evaluation. There may be a good idea here but I cannot make sense of it in the paper's current state.

---

### Official Review · Reviewer_kWNG · 2026-05-22
**This paper introduces a new framework called H-TDBU (Hierarchical Hybrid Top-Down and Bottom-Up) for generating high-quality synthetic tabular data.**

**Rating:** 5
**Confidence:** 4

**Review:**

**Strengths**
- its interesting that instead of treating data generation as a single mathematical problem, H-TDBU splits the task into two paths to decouple semantic structure from stochastic texture.
- simple methodology where semantics are first generated and used as top-down whereas bottom-up independently learns statistical patterns. Then during the reconciliation stage, both the logic skeleton S and the latent texture Z are fed into the hybrid top-down/bottom-up generator.
- good results

**Areas of Improvement**
- Synthesis and Reconciliation is unclear to me, how exactly is the reconciliation performed? Lacks sufficient implementation details for reproducibility.
- "struggling with data heterogeneity, logical consistency, rare-event coverage, and robustness in low-data regimes" -- elaborate on this in future work?